# Micronutrients and Major Depression: A Mendelian Randomisation Study

**DOI:** 10.3390/nu16213690

**Published:** 2024-10-29

**Authors:** Rebecca E. Carnegie, Jie Zheng, Maria C. Borges, Hannah J. Jones, Kaitlin H. Wade, Hannah M. Sallis, Sarah J. Lewis, David M. Evans, Joana A. Revez, Jonathan Evans, Richard M. Martin

**Affiliations:** 1Centre for Academic Mental Health, Population Health Sciences, Bristol Medical School, University of Bristol, Bristol BS8 2BN, UK; j.evans@bristol.ac.uk (J.E.); 2Medical Research Centre (MRC), Integrative Epidemiology Unit (IEU), University of Bristol, Bristol BS8 1QU, UK; 3Population Health Sciences, Bristol Medical School, University of Bristol, Bristol BS8 1QU, UK; 4Department of Endocrine and Metabolic Diseases, Shanghai Institute of Endocrine and Metabolic Diseases, Ruijin Hospital, Shanghai Jiao Tong University School of Medicine, Shanghai 200025, China; 5Shanghai National Clinical Research Center for Metabolic Diseases, Key Laboratory for Endocrine and Metabolic Diseases of the National Health Commission of the PR China, Shanghai National Center for Translational Medicine, Ruijin Hospital, Shanghai Jiao Tong University School of Medicine, Shanghai 200025, China; 6NIHR Biomedical Research Centre, a Partnership between University Hospitals Bristol and Weston NHS Foundation Trust and the University of Bristol, Oakfield House, Oakfield Grove, Bristol BS8 2BN, UK; 7School of Psychological Science, University of Bristol, Bristol BS8 1TU, UK; 8The University of Queensland Diamantina Institute, The University of Queensland, Brisbane, QLD 4102, Australia; 9Institute for Molecular Biosciences, The University of Queensland, Brisbane, QLD 4072, Australia

**Keywords:** mendelian randomization, vitamins, minerals, micronutrients, major depressive disorder

## Abstract

Background: Various vitamins and minerals have been implicated in the aetiology of depression. Objective: To estimate the effects of micronutrient exposures on major depressive disorder (MDD) and recurrent depression (rMDD) using Mendelian randomisation (MR), a method using genetic data to estimate causal effects given certain assumptions. Methods: We undertook a comprehensive bidirectional MR study of multiple micronutrient exposures on MDD and rMDD. Summary statistics were obtained from the Psychiatric Genomics Consortium (PGC) genome-wide association studies (GWASs) of MDD (cases = 116,209; controls = 314,566) and rMDD (cases = 17,451; controls = 62,482). Results: None of the micronutrients with available genetic instruments were strongly associated with MDD or rMDD using traditional MR methods. However, using methods to increase analytical power by accounting for genetically correlated variants (e.g., cIVW) highlighted five micronutrients with possible causal effects. Point estimates for rMDD were the largest magnitude, with three micronutrients suggestive of a protective effect: serum iron (OR_cIVW_ 0.90 per SD increase; 95% CI 0.85–0.95; *p* = 0.0003); erythrocyte copper (OR_cIVW_ 0.97; 95% CI 0.95–0.99; *p* = 0.0004); and 25(OH) vitamin D (OR_cIVW_ 0.81; 0.66–0.99; *p* = 0.04). Apparent adverse effects of increased selenium on the risk of MDD (OR_cIVW_ 1.03; 95% CI 1.02–1.05; *p* = 0.0003) and rMDD (OR_cIVW_ 1.08; 95% CI 1.00–1.08; *p* = 0.06), and serum magnesium on rMDD (OR_cIVW_ 1.21; 1.01–1.44; *p* = 0.04); were less consistent between methods and may be driven by pleiotropy. Conclusions: Our results suggest weak evidence for a protective effect of iron, copper and 25(OH)D on major depressive outcomes, with mixed evidence for selenium and magnesium. There was no evidence to support a causal effect of any other micronutrients on MDD or rMDD, although genetic instruments were lacking, with insufficient power to detect small but important effects. Future micronutrient supplementation trials should ensure ample statistical power given modest causal effect estimates and consider potential risks of supplementation, as some micronutrient effect estimates suggested potential harm in excess.

## 1. Introduction

Vitamins and minerals, collectively categorised as ‘micronutrients’, are essential for human health and optimal physiological functioning [1]. Micronutrient deficiencies have been associated with multiple pathological processes, with characteristic micronutrient deficiency syndromes resulting from inadequate intake. ‘Dietary reference intakes’ for specific micronutrients aim to encourage adequate dietary intake at a population level [1]. However, the decline of whole food diets in favour of energy-dense but nutritionally-replete processed foods may underlie more widespread subclinical micronutrient deficiency and depletion, with unclear repercussions for population health [1,2,3,4,5].

Major depressive disorder (MDD) is a leading cause of global morbidity, with recent estimates that 1 in 2 people will be affected in their lifetime [6]. Various micronutrients have been implicated in the aetiology of depression over the past few decades [4], based on findings from human observational [7,8,9,10] and animal [11,12,13] studies. Arguments in support of a causal role for micronutrients in depression are founded on direct physiological effects [4,14], animal studies of micronutrient deprivation [11,12,13], and overlapping symptoms between micronutrient deficiency and depression [4]. It is possible that subclinical micronutrient deficiencies may contribute to the burden of depressive disorders at the population level, but it is difficult to rule out the possibility that associations observed between micronutrient status and depression are the result of confounding or reverse causality and compounded by publication bias.

Conducting well-designed randomised controlled trials (RCTs) of micronutrient supplementation is expensive and time-consuming. Existing evidence suggests that individual micronutrient effect sizes are likely to be small, necessitating large sample sizes, especially for prevention trials. Prior RCTs of micronutrient supplements for Major Depressive Disorder (MDD) have shown mixed results [15], and the choice of which micronutrients, doses, durations, combinations and populations is problematic. Nutraceutical depression prevention trials are uncommon, with two major trials in the literature finding no evidence that micronutrient supplementation reduces the risk of depressive disorder [16,17]. The European Mood Food Trial nutraceuticals arm suggested no benefit of a mixed micronutrient supplement (combining selenium, folic acid, vitamin D and omega 3) for reducing depression onset among 1025 overweight and obese participants (Odds ratio (OR) 1.06 (95% CI, 0.87–1.29)) [16]. Similarly, no evidence of benefit was identified for vitamin D supplementation in the Vitamin D and Omega-3 Trial-Depression Endpoint Prevention (VITAL-DEP) trial [17], which randomised 18,353 non-depressed individuals over 50 s to receive vitamin D or placebo over 5 years (hazard ratio 0.97; (95% CI, 0.87–1.09); *p*  =  0.62). Further understanding of the likelihood and estimated magnitude of possible causal effects may help inform micronutrient and participant selection and estimate statistical power to improve the efficiency of intervention development.

Mendelian randomisation (MR) is a causal analysis method which uses germline genetic variants (usually single nucleotide polymorphisms, SNPs) as ‘instrumental variables’ to proxy potentially modifiable risk factors [18,19]. MR aims to strengthen the evidence for causality by exploiting the random nature of allele segregation at meiosis and allocation at conception to reduce the impact of confounding and reverse causation seen in conventional epidemiology [18]. Two-sample MR uses summary statistics from separate genome-wide association studies (GWASs) of exposures (sample 1) and outcomes (sample 2), which increases the availability and sample size of available data. Causal estimates derived from MR can increase research efficiency by prioritising scarce resources for the development of interventions supported by robust causal evidence.

Several studies have used MR to investigate nutritional factors in the aetiology of depression, implicating both carbohydrates [20] and lipid metabolic pathways [21]. Evidence to suggest a causal link between micronutrients and depression is less convincing. Studies of B vitamins and depression have found no association with either vitamin B12 (OR per SD 0.96; 95% CI 0.52–1.79; *p* = 0.91) [22], folate (OR 1.18; 95% CI 0.18–7.66; *p* = 0.86) [22], or homocysteine (OR 0.95; 95% CI 0.88–1.00; *p* = 0.12) [22], although the outcome sample sizes may have been too small to detect effects (see Appendix A). The MR-estimated effects of multiple minerals in a sample of 10,640 Chinese women found no link with serum calcium (OR per SD 0.92; 95% CI 0.67–1.28; *p* = 0.63), magnesium (OR per SD 1.19; 95% CI 0.22–6.61; *p* = 0.84), iron (OR per SD 0.98; 95% 0.91–1.05; *p* = 0.60) or zinc (OR per SD 0.99; 95% CI 0.95–1.03; *p* = 0.66) on the risk of depression [23]. However, in addition to the relatively small sample size, results may have been affected by the mix of ancestry between exposure and outcome GWASs. Multiple MR studies to date have looked at the association of 25-hydroxyvitamin D (25(OH)-vitamin D) and depression, with increasing sample sizes and methodological complexity [24,25,26,27,28]. These studies show consistent effect estimates suggesting a small protective effect of vitamin D on depression (~2–3% reduction in risk per SD increase in vitamin D for two sample analyses and ~6% for one sample MR), although confidence intervals overlapped the null for most estimates. The power to detect small effects is increasing with GWAS sample sizes. The most recent vitamin D MR (a supplementary analysis within a UKBB vitamin D GWAS) [28] suggested weak evidence for a small causal effect of vitamin D on MDD (odds ratio (OR) per SD (equivalent to 21.0 nM/L 25(OH)D): 0.98; 95% CI 0.96–1.00; *p* = 0.03). However, this did not pass the threshold of multiple testing or replication across MR methods. There was also stronger evidence for the reverse effect, in which genetic liability to MDD reduced 25(OH) vitamin D (−0.04 (95% CI −0.01–−0.07); *p* = 0.005), which may have driven an apparent effect. Sample overlap between exposure and outcome data in this analysis, along with high heterogeneity due to the multitude of instruments with variable clarity about their role in the physiology of vitamin D, makes further validation important.

We are not aware of any micronutrient MR studies to date using recurrent MDD (rMDD) as an outcome, representing a novel opportunity to investigate the more chronic and severe end of the depressive disorder spectrum and theoretically investigate the evidence for a ‘dose-response’ relationship between micronutrients and depression. Given that many micronutrients have some evidence linking them to depression through symptom overlap of deficiency states, observational associations with depression, or a plausible biological pathway to depression, we undertook a two-sample bidirectional MR study of fourteen micronutrient exposures with available genetic instruments on both MDD and rMDD outcomes. By considering multiple micronutrients, we aimed to highlight potential micronutrients for further translational research in MDD and provide a framework for similar micronutrient MR studies for other outcomes. Some recent MR studies have used a similar approach to investigate the impact of multiple micronutrients on cancer [29,30], amyotrophic lateral sclerosis [31], and COVID-19 severity [32], theoretically improving research efficiency and reducing the risk of publication bias.

## 2. Materials and Methods

### 2.1. Data Sources

#### 2.1.1. Micronutrient Exposures

For this analysis, we defined micronutrients as vitamins and minerals, which we categorised into four groups: water-soluble vitamins, fat-soluble vitamins, macrominerals and microminerals or trace elements (Figure 1). We searched the GWAS catalogue for GWAS studies of 31 vitamins and minerals in European populations to identify genetic variants that could be used as proxies for micronutrient status in the MR analyses (Figure 2, Table 1). Micronutrients with no existing GWAS among European ancestry cohorts or lack of strongly associated SNPs (*p* < 5 × 10^−8^) were excluded from the analyses (Figure 2). Circulating levels of micronutrients were chosen where possible, though other measurements (e.g., measures of urinary excretion) were considered where no SNPs robustly associated with serum levels were identified. 

SNPs were selected as genetic instruments if they reached the traditional genome-wide association *p*-value thresholds in the relevant GWAS (*p* < 5 × 10^−8^). Where necessary, units of the SNP-exposure association were converted to standard deviation (SD) units by dividing the beta and standard error by the standard deviation (SD) of the exposure.

Where genetic instruments for exposures had been replicated in an independent sample, we utilised the effect estimate from the two-stage GWAS meta-analysis where available. Several of the micronutrient-associated SNPs had not been replicated in an independent cohort, potentially leading to an overestimate of the SNP-exposure association (i.e., “winner’s curse”) [49]. This potentially limited findings for copper, zinc, manganese, vitamin B6 and homocysteine (Table 1). Where possible, we obtained full summary statistics for the GWASs, either through publicly accessible data repositories or by contacting the corresponding author of the study directly.

Several micronutrients had GWAS studies of more than one measure of physiological status, and separate analyses were conducted for each available measure. Plasma homocysteine was used in addition to plasma folate, as it is a sensitive marker of vitamin B9 deficiency and is thought to be on the pathway to various adverse health outcomes. Beta carotene and serum retinol were both used for vitamin A status, and ferritin was used, in addition to serum iron, as a marker of iron stores.

As the largest GWAS of 25(OH) vitamin D was undertaken in UKBB (N = 417,580)–the same population as our rMDD outcome [28]. We chose instruments from an earlier GWAS analysis of 79,366 participants from the SUNLIGHT consortium to avoid biasing estimates towards the observational association due to sample overlap [39]. The SUNLIGHT GWAS had the advantage of not containing cohort studies common to either outcome sample, and the genetic variants associated with 25(OH)D are replicated in other samples.

#### 2.1.2. Depression Data

The PGC is an international consortium of studies with genotype data from over 900,000 individuals and 38 countries. The PGC MDD sample used in this study contains genotype data from 246,363 cases and 561,190 controls (Appendix A) [50,51,52]. Summary statistics from this cohort were obtained to provide two distinct outcome samples. For our primary outcome, MDD, we used summary statistics from the PGC with UK Biobank participants (UKBB) removed (n = 430,775, cases = 116,209; controls = 314,566) [50]. Studies contributing data to the PGC MDD GWAS, including diagnostic criteria, inclusion, and exclusion criteria, are detailed in Appendix A. The MDD GWAS used as our primary outcome combines studies with clinically defined diagnostic criteria (referred to as PGC_139k) [52] with genetic data from 23andme, which relies on participant self-report [50]. However, a strong genetic correlation exists between the clinically defined and 23andme data (r_G_ = 0.85 (se = 0.03.) As a secondary outcome, we used rMDD, which was derived exclusively from participants in UKBB [34]. The rMDD summary statistics were taken from a PGC GWAS confined to the UK Biobank (UKBB) cohort, based on DSM-5 diagnoses from an online mental health questionnaire (N = 80,933; cases = 17,451; controls = 62,482) [34]. Individuals self-reporting two or more lifetime depressive episodes across their lifetime were classified as having rMDD (n = 17,451), and controls were those with no prior episodes of depression (n = 63,482) [34]. In addition to checking the validity of results across distinct outcomes, using rMDD as a secondary outcome enabled us to refine the MDD phenotype to a more chronic and pervasive depression.

### 2.2. Statistical Power

Using the estimated proportion of variance (R^2^) in the level of each micronutrient explained by our genetic instruments and the F-statistics for each exposure (Appendix A), we estimated the power for our analyses using an online power calculator for MR studies (https://shiny.cnsgenomics.com/mRnd/ accessed on 15 January 2024) [53] assuming a = 0.05. As this online calculator is designed for use in one-sample MR studies, power estimates are provided as an approximation only. We estimated that we had over 80% power to detect a moderate causal effect (OR ≥ 1.1 per SD in each micronutrient) for 12 micronutrients (Table 1). Exceptions to this were serum folate (power = 62%), vitamin E (46%), calcium (74%), sodium (24%) and potassium (power = 9%). Micronutrients with power < 60% were excluded from analyses (see Figure 2.)

### 2.3. Statistical Analysis

Analysis was undertaken in R 4.0.2 [54] using the TwoSampleMR analysis package (version 0.5.6) [55] and the MendelianRandomization package (version 0.5.1) [56]. Results showing the MR-estimated effect of micronutrients on MDD and rMDD were plotted using the ggplot2 R package 3.5.1 [57].

The TwoSampleMR package (version 3.5.1) [55] was used to harmonise the exposure and outcome data, clump the instrument based on LDs (r^2^. threshold < 0.001) and calculate MR-derived effect estimates. For exposures with a single SNP, we derived Wald ratio estimates using the SNP outcome and SNP-exposure data from the corresponding GWAS. Where more than one SNP was available, we used the inverse variance weighted (IVW) method to pool individual Wald ratio estimates for each SNP. For micronutrients with three or more SNPs, we also used the MR-Egger method to detect and account for directional pleiotropy by relaxing the assumption of no unbalanced pleiotropy for the IVW methods. MR estimates using weighted median and weighted mode methods were included in sensitivity plots to check consistency across MR methods. Full details of the SNPs included in analyses are given in Appendix A.

To investigate heterogeneity between instruments on the risk of MDD and rMDD, we calculated Cochran’s and Rucker’s Q values (see Appendix A). For analyses with high heterogeneity, the MR-Egger intercept term was used to check whether SNP heterogeneity is directionally biased MR estimates. Scatter plots, forest plots, leave-one-out and funnel plots were graphed for individual micronutrients for both MDD and rMDD (Appendix A).

Although we prefer to consider the overall strength of evidence instead of *p*-value thresholds, for the purpose of further interpretation, we considered that a Bonferroni corrected *p*-value < 0.004 would strongly support an association. This is likely to be an overly stringent measure, assuming a = 0.05, divided by 14 individual tests for which data was available (Figure 1 and detailed below). However, micronutrients with *p* < 0.05 and consistent effects across methods and outcomes are also discussed.

#### 2.3.1. Functional MR

Where possible, we repeated MR analyses after restricting the instruments to only those with clear physiological relevance for each exposure (denoted “functional” analyses). Details of gene function were obtained from the National Library of Medicine (NLM) Gene database (https://www.ncbi.nlm.nih.gov/gene/ accessed on 4 March 2024). Genes encoding proteins directly related to the metabolism of each micronutrient (defined as processes related to micronutrient uptake, synthesis, transport, or degradation) were included in “functional” sensitivity analyses. Where the description given on the NLM Gene database did not clearly specify a physiological function with direct relevance to the metabolism of the specified micronutrient, biological pathways for the gene were cross-checked with the Kyoto Encyclopedia of Genes and Genomes (KEGG) pathway resource (https://www.kegg.jp/kegg/ accessed on 4 March 2024) [58].

#### 2.3.2. Correlated MR

As several exposure GWASs had identified only a handful of independent SNPs, we used the generalised MR methods suggested by Burgess et al. [59,60] to maximise analytical power where full summary statistics were available. As this method accounts for correlation between SNPs using an LD matrix of r^2^ values, it facilitates retention of SNPs that would otherwise be excluded due to traditional LD clumping thresholds, thus maximising power without overestimating precision of the estimated effect (resulting in overly narrow confidence intervals) [59]. To achieve this, we used the MendelianRandomization R package [56], which accounts for the correlation between SNPs in our instrument sets. For these analyses, we used a more relaxed LD clumping threshold (r^2^ < 0.2) to prune only SNPs in very high LD. For this paper, the acronym cIVW has been used to differentiate this from traditional IVW, which does not account for the correlation between SNPs using an LD matrix. The LD matrix panel was derived from individual phenotype data from a random selection of 10,000 UK Biobank participants [61].

#### 2.3.3. Multivariable MR

As micronutrient exposures are highly physiologically correlated, we considered the use of multivariable MR (MVMR) [62] where appropriate and where full summary statistics allowed, using the MVMR ‘R’ package [63]. The MVMR R package was used to calculate conditional F-statistics, assuming genetic covariance to be zero due to non-overlapping exposure samples. Due to data availability, we were limited to a single MVMR model, accounting for the effects of 25 (OH) vitamin D, calcium and magnesium metabolism on the risk of depression simultaneously. These micronutrients are highly physiologically interconnected [64], and yet their effects on the aetiology of depression may be directionally opposing.

#### 2.3.4. Reverse MR

We used two methods to investigate the possibility of reverse causality (i.e., where genetic liability to MDD could drive an apparent MR association). Firstly, we applied Steiger filtering to remove SNPs that explained a greater degree of variance in the outcome than the exposure to reduce the chances that the observed effects were being driven by a primary effect of the SNP on the outcome rather than exposure.

Secondly, where full micronutrient summary statistics were available, we undertook reverse MR, using MDD as the exposure and micronutrient summary statistics as the outcome. Reverse MR was undertaken using genome-wide significant SNPs from the complete MDD GWAS [50]. (n = 807,553). MR in the reverse direction was undertaken using the same clumping thresholds as our primary analyses (r^2^ < 0.001) for traditional IVW analyses (nSNPs = 66) and (r^2^ < 0.2) for cIVW (nSNPs = 110). For these analyses, effect estimates represent the SD change in each micronutrient with an approximate doubling of the genetic liability to depression.

## 3. Results

### 3.1. Effect of Micronutrients on Depressive Disorders

Results of MR analyses are summarised in Table 2 and Figure 3. Heterogeneity statistics are presented in Appendix A, and plots summarising individual micronutrient analyses are presented in Appendix A.

OR_IVW_ analyses are presented for all micronutrients with more than one associated SNP, and OR_wald_ is presented for micronutrients with single SNP instruments. Results for MR-Egger are also presented for micronutrients with 3 or more associated SNPs. The results of analyses when restricting to SNPs with clearly defined functional roles in micronutrient metabolism are also given, defined as ‘functional’ in the method, to differentiate from estimates derived from IVW or Wald estimates with a less discriminate set of instruments. Results for correlated analyses (cIVW and cEgger) are presented in Table 3. Heterogeneity statistics are given in Appendix A.

MR estimates are given as ORs per standard deviation increase (95% CI) for MDD and rMDD outcomes. Figure 3a summarises IVW estimates for each available micronutrient (or Wald ratio for single SNP analyses) and for ‘functional’ analyses (i.e., including only SNPs with clear biological function). As all SNPs for vitamin A, vitamin B6, and vitamin B9 (folate) had a clear biological function, functional analyses are equivalent to IVW or Wald estimates. Figure 3b MR estimated OR for depression comparing traditional IVW (clumping thresholds r^2^ < 0.001) with results from correlated methods (cIVW and cEgger), in which clumping thresholds have been increased(r^2^ < 0.2), and MVMR estimates for magnesium and vitamin D. cIVW and cEgger account for LD-correlation in the methods, and so preserve more SNPs for analyses, but require full summary statistics are available, (in this case vitamin D, magnesium iron, ferritin, selenium, zinc and copper.) Copper, Selenium and Zinc lack MR Egger estimates due to insufficient SNPs after LD clumping.

Using traditional MR methods, effect sizes for all micronutrients on MDD were small, with confidence intervals close to or overlapping the null. MR effect estimates for our secondary outcome, rMDD, were similarly modest in magnitude and lacking consistency, with wider confidence intervals in keeping with the smaller sample size. None of the estimates reached the Bonferroni corrected *p*-value threshold (*p* < 0.004) using our primary methods.

Steiger filtering did not remove any SNPs from the analyses, as all genetic instruments explained more variance in the micronutrient exposure than MDD and rMDD outcomes, meaning results were unchanged.

The availability of full genetic summary statistics for six micronutrients (i.e., Vitamin D, Magnesium, Iron, Copper, Selenium and Zinc) facilitated the use of correlated MR methods (i.e., cIVW and cEgger), which increased analytical power by using additional genetic variants in partial LD (r^2^ > 0.2). Using correlated methods, we identified possible protective effects for increased genetically predicted vitamin D, copper, and iron on the risk of depression, with some inconsistent evidence suggesting an increased risk of depression with increasing serum magnesium and selenium. MR results for these five micronutrients are summarised below, with details in Table 3 and Figure 3b. A table comparing standard errors (SE) between correlated (SE_cIVW_) and traditional IVW analyses (SE_IVW_) is provided in Appendix A.

#### 3.1.1. Vitamin D

Although MR estimates for the effect of 25(OH) vitamin D on MDD suggested no effect (OR_cIVW_ 0.99; 95% CI 0.93, 1.06; *p* = 0.86) in keeping with traditional MR methods, there was evidence for heterogeneity (Cochran’s Q 44; *p* = 0.03), with directional bias suggested by the Egger intercept (1.01; *p* < 0.001.) The correlated Egger analysis suggested strong evidence for an effect of 25(OH) vitamin D on MDD (OR_cEgger_ 0.86 per SD increase; 95% CI 0.78–0.95; *p* = 0.003), which was consistent with rMDD effect estimates (OR_cIVW_ 0.87; 95% CI 0.74, 1.02; *p* = 0.08; and OR_cEgger_ 0.81; 95% CI 0.64, 1.03; *p* = 0.13), for which heterogeneity was low (Cochran’s Q 35; *p* = 0.16; Rucker’s Q 35; *p* = 0.14; and Egger Intercept 1.00; *p* = 0.47; see Appendix A.) Effect estimates restricted to four SNPs with clear physiological function were consistent in both magnitude and direction for rMDD (OR_Functional_ 0.89; 0.77, 1.03; *p* = 0.10), with MDD estimates reflective of IVW and cIVW estimates (OR_Functional_ 0.99; 0.95, 1.03; *p* = 0.52). MVMR models accounting for genetically correlated effects of magnesium and calcium homeostasis were consistent with univariable estimates for both outcomes (MDD OR_MVMR_ 0.99 per SD; 95% CI: 0.86–1.15; *p* = 0.92; and rMDD OR_MVMR_ 0.75 per SD (95% CI 0.50–1.12 *p* = 0.16), Table 4).

#### 3.1.2. Magnesium

There was no evidence of an effect of serum magnesium on MDD, with inconsistent estimates between methods. Conversely, MR estimated effects were directionally consistent with an adverse effect of serum magnesium on rMDD risk across all methods. Confidence intervals for MR analyses using traditional methods all included the possibility of no effect, except when restricting the instruments to a single ‘functional’ variant (OR_wald_ 1.38 per SD of genetically predicted serum magnesium (~0.1 mmol/L); 95%CI 1.07–1.78; *p* = 0.01) The direction of MR estimates for correlated analyses were consistent with an adverse effect of serum magnesium on rMDD (OR_cIVW_ 1.12; 95% CI 0.95, 1.33; *p* = 0.19; OR_cEgger_ 1.61; 95% CI 0.97–2.68; *p* = 0.07,) and with traditional estimates using all 5 SNPs (OR_IVW_ 1.14; 95%CI 0.92–1.41; *p* = 0.23; and OR_Egger_ 1.62; 95% CI 0.94, 2.77; *p* = 0.18.) However, using MVMR to account for physiologically correlated genetic effects of magnesium, vitamin D and calcium, the direction of the effect of serum magnesium on rMDD was reversed (OR_MVMR_ 0.88 (95% CI 0.70–1.09) *p* = 0.26, Table 4), potentially highlighting pleiotropic effects through 25 (OH) vitamin D and calcium homeostasis.

#### 3.1.3. Copper

As only two copper SNPs were available for MR when using traditional clumping thresholds, and neither had a clear biological function, traditional MR methods were limited. IVW effect estimates suggested little evidence for an association between copper and either depression outcome (MDD: OR_IVW_0.98 per SD increase (~60 μg/L); 95% CI 0.95–1.01, *p* = 0.19; rMDD: OR_IVW_0.96; 95% CI 0.86–1.07, *p* = 0.46.) However, correlated MR tightened the confidence intervals around the estimates, suggesting a very small inverse association between erythrocyte copper and both depression outcomes when using seven SNPs; MDD OR_cIVW_0.98; (95% CI 0.97–0.99, *p* = 0.0003) and rMDD OR_cIVW_0.97; (95% CI 0.95–0.99, *p* = 0.002). Point estimates for MDD were consistent across all available methods (IVW, cIVW and cEgger), while for rMDD the cEgger effect was null (OR_cEgger_ 1.00; 95% CI 0.91, 1.10; *p* = 1.00), although SNP heterogeneity was low (Q = 4; *p* = 0.64), with no evidence for directional pleiotropic bias (Egger Intercept 0.98; *p* = 0.46). Cautious interpretation of results is recommended, particularly due to the lack of clearly defined biology for the instruments and GWAS replication for these instruments, but also due to the increased precision from the two SNP analyses (SE_IVW_/SE_cIVW_ ratios 2.63 for MDD and 3.06 for rMDD, see Appendix A).

#### 3.1.4. Iron

Correlated MR suggested weak evidence for a protective effect of serum iron on MDD (OR_cIVW_ 0.97; 95% CI 0.94–1.00; *p* = 0.03), with the magnitude of effect consistent across methods. SNP heterogeneity was high for MDD analyses (Q= 101; *p* < 0.001 and Rucker’s Q = 101; *p* = 7.5 × 10^−10^) but was assumed to be balanced (Egger intercept =1; *p* = 0.79.)

Traditional MR suggested weak evidence for an inverse association between four SNPs for serum iron and rMDD (OR_IVW_ 0.90 per SD; 95% CI 0.81–1.00; *p* = 0.05), with confidence intervals tightened by using 31 SNPs and correlated MR (OR_cIVW_ 0.87; 95% CI 0.83–0.91; *p* = 3.2 × 10^−9^). All four SNPs used in traditional analyses for serum iron were identified as having a clear biological function, which further strengthened evidence estimates for serum ferritin were all directionally consistent with adverse effects of iron deficiency on depression outcomes, though confidence intervals were wide (see Table 3).

#### 3.1.5. Selenium

A small increased risk of MDD was identified for genetically predicted serum selenium using correlated MR (OR_cIVW_ per SD increase (~220 μg/L) 1.03; 95% confidence intervals (CI): 1.02–1.05; *p* = 0.001.) However, Egger estimates were directionally reversed (OR_cEgger_ 0.94, 95% CI 0.84–1.05) *p* = 0.29), suggesting potential pleiotropy. Similar inconsistencies were found between estimates for rMDD, with point estimates suggesting an increased risk of rMDD using traditional MR methods (OR_wald_ 1.14; 95% CI 0.97–1.34; *p* = 0.12) and cIVW analyses (OR_cIVW_ 1.08; 95% CI: 1.02–1.15; *p* = 0.01), but not cEgger analyses (OR_cEgger_ 0.97; 95% CI 0.67–1.39; *p* = 0.86). Neither Egger intercept test suggested that the estimates might be biased by directional pleiotropy (MDD cEgger intercept = 1.02; *p* = 0.10; rMDD cEgger intercept = 1.03; *p* = 0.54). Cautious interpretation of these results is needed, as in addition to inconsistent estimates between methods, the relative precision of the correlated estimates compared to the Wald ratio is worthy of note (SE_IVW_ /SE_cIVW_ ratios 2.32 for MDD and 2.63 for rMDD, Appendix A) Finally, genetic instruments for serum selenium lacked well-defined metabolic functions, preventing MR analyses restricted to functional variants, which may further limit interpretation.

### 3.2. Effect of Major Depressive Disorder on Micronutrient Status

Results for reverse MR analyses (i.e., where MDD was considered the exposure) for vitamin D, copper, magnesium, iron, and selenium are presented in Table 5. In our primary analyses, there was little evidence to suggest that genetic liability to depression altered micronutrient status. Each doubling of genetic tendency to depression reduced 25(OH) vitamin D by 0.02 SD (beta_IVW_ −0.02; 95% CI −0.05, 0.01; *p* = 0.15) and increased trace elements by between 0.01 and 0.04 SD: selenium (beta_IVW_ 0.01; 95% CI −0.29, 0.31; *p* = 0.81); copper (beta_IVW_ 0.04; 95% CI −0.24, 0.32; *p* = 0.77) and iron (beta_IVW_ 0.02; 95% CI −0.08, 0.12; *p* = 0.65). However, using cIVW, there was a weak suggestion that genetic susceptibility to depression lowered 25(OH) vitamin D (beta_cIVW_ −0.02; 95% CI −0.04, 0.00; *p* = 0.09) and increased selenium (beta_cIVW_ 0.27; 95% CI 0.01, 0.52; *p* = 0.04), although these results were attenuated after Steiger filtering, (25(OH)vitamin D beta_cIVW_: 0.00 (95% CI −0.02–0.02; *p* = 0.72); selenium beta_cIVW_ 0.00 (95% CI −0.58, 0.58) *p* = 0.87). Steiger filtering removed a significant number of MDD SNPs for each micronutrient (13 SNPs for vitamin D, 51 for copper, 18 for iron, and 48 for selenium), suggesting that multiple SNPs used as MDD instruments explained a greater proportion of the variance in the micronutrient exposure than MDD. Steiger filtering also resulted in marked improvements in heterogeneity in the reverse analyses, with minimal change to the estimated variance explained (Appendix A).

## 4. Discussion

To our knowledge, this is the first comprehensive MR study assessing the likely causal effect of multiple micronutrients in the aetiology of major depression. None of the micronutrients had resoundingly conclusive results, with small effect sizes, imprecise estimates, and inconsistencies between methods and outcomes. In analyses with the greatest statistical power, which included and accounted for correlated genetic instruments, we found some evidence to suggest that 25(OH) vitamin D, serum iron, and possibly copper had protective effects on depressive illness. The greatest magnitudes of effect were seen for rMDD, conceptually supporting the notion of a dose-response relationship, although estimate precision was hindered by the relative sample size. Each SD increase in serum iron reduced the risk of rMDD by approximately 10% (95%CI 5–15%), 25(OH) vitamin D by ~19% (1–34%), and copper by ~3% (95% CI 1–5%), reflecting potentially important effects worthy of further exploration as genetic samples and instruments evolve. Several of the MR estimates for serum magnesium and selenium suggested an increased risk of depressive outcomes, particularly using MR methods accounting for genetic correlation. However, cautious interpretation is recommended due to inconsistencies between methods (specifically, cEgger estimates for selenium and MVMR for magnesium), suggesting potential pleiotropy. We found no evidence to suggest that our results were due to the reverse causal effect of MDD on micronutrient status.

Vitamin D has been the most thoroughly investigated of all micronutrients in depression, with inverse associations in observational studies [64,65,66,67], although mixed results among trials [68,69]. Evidence of causality has been further undermined by prior MR studies [24,25,26,27,28], using increasingly powerful sample sizes and genetic instruments. Our findings in MDD are comparable to the effect sizes seen in previous MR analyses, with the latest finding weak evidence for an effect of 25(OH) vitamin D on MDD, that did not pass the threshold of multiple testing or replicate across MR methods and was potentially driven by the effect of MDD on 25(OH) vitamin D [28]. One potential limitation of this analysis, however, is the large number of SNPs with uncertain relevance to vitamin D physiology, which may have diluted a small effect and increased the potential for horizontal pleiotropy. Using a small number of well-validated and categorised SNPs and including analyses restricted to functional variants reduced the heterogeneity in our analyses. Furthermore, minimising sample overlap between exposure and outcome samples increases confidence in the results. Our effect estimates were remarkably consistent across methods, with larger point estimates in rMDD and again when accounting for the potentially pleiotropic effects of calcium and magnesium homeostasis using MVMR. Once Steiger filtering had removed SNPs explaining a greater variance in 25(OH) vitamin D, we found no evidence that MDD lowered 25(OH) vitamin D using reverse MR. As far as we are aware, no MR studies have considered the effect of 25(OH) vitamin D on the risk of recurrent depression. Although power for rMDD analyses was weaker, an approximate ~13% reduction in risk per SD (OR_cIVW_ 0.87 95% CI 0.74–1.02; *p* = 0.08) may support trials into secondary prevention, with the effect estimate able to guide power and sample size calculations.

MR evidence for iron in our main analyses was weak. However, estimates using cIVW suggested a 13% decrease in the odds of rMDD per SD of serum iron (OR_cIVW_ 0.87; 95% CI 0.83–0.91; *p* = 3.2 × 10^−9^). Results were consistent across methods and between outcomes. Along with reciprocal changes in serum ferritin, the results are compatible with the hypothesis of an adverse effect of iron deficiency on the risk of depression, especially for recurrent MDD. We found no evidence that this was driven by reverse association—in which people at risk of depression have reduced iron or ferritin, which might be the case among those with poor dietary intake. There is a clear symptom overlap between iron deficiency and depression, particularly fatigue and weakness, with evidence from observational studies suggesting an increased risk of depression among those with iron deficiency anaemia, reduced by supplementation [70,71]. Iron deficiency has been linked to reductions in dopamine receptors in animal studies, suggesting that this finding may go beyond symptom overlap [72]. Further work could consider mediating mechanisms for this association, as well as consider whether this estimate could be masking a more substantial non-linear association, with both iron deficiency and excess having adverse effects.

Several MR methods estimated increased risk of recurrent depression from genetically predicted serum magnesium levels, particularly when restricting instruments to those with clearly defined roles in magnesium homeostasis OR_wald_ 1.38 per SD of serum magnesium (95% CI 1.07–1.78; *p* = 0.01) and using correlated MR methods (OR_cIVW_ 1.12; 95% CI 0.95–1.33; *p* = 0.19 for MDD; and OR_cIVW_ 1.61; 95% CI 0.97–2.68; *p* = 0.07 for rMDD). Genes linked to magnesium ion binding and transport mechanisms have been identified in a PGC GWAS of multiple psychiatric outcomes [73], while a prior MR study of multiple minerals identified a significantly increased odds of bipolar disorder from serum magnesium (OR 8.78 per SD increase; 95% CI 1.16–66.26; *p* = 0.04) [23]. MR estimates for magnesium in MDD were smaller and inconsistent, possibly reflecting greater phenotypic overlap between rMDD and bipolar disorder. The effect direction for magnesium appears in contrast with observational and trial data, suggesting the benefits of magnesium for mood disorders [74,75]. This discrepancy may highlight the limitations of using serum magnesium as a reflection of magnesium status, as it is known to be poorly correlated with intracellular stores [76]. Effect estimates for serum magnesium may be unable to capture micronutrient effects at a cellular and tissue level, especially within the brain. Furthermore, it is important to note that the effect of serum magnesium on rMDD was reversed when accounting for genetically correlated effects of 25(OH) vitamin D and calcium using MVMR (OR_MVMR_ 0.88; 95% CI 0.70–1.09; *p* = 0.26), suggesting possible pleiotropy. The physiology of calcium, magnesium and vitamin D are intricately linked and highly complex, and each micronutrient may have differential effects on MDD–as suggested by trials supplementing combined vitamin D and calcium appearing less effective than vitamin D alone [77]. The single genetic variant used in functional analyses on the TRPM6 gene is primarily involved in magnesium transport across the intestinal lumen and distal convoluted tubule, with variations associated with hypomagnesaemia and secondary hypocalcaemia [78]. Further work to establish the biological underpinnings of these results is therefore essential, as interventions to correct excess body magnesium may be in direct conflict with those addressing inadequate cellular magnesium uptake.

Although adverse effects of excess selenium have been previously identified using MR [79] and in randomised trials [80], to our knowledge, this is the first MR study to investigate effects on MDD. Methods to account for the correlation between SNPs suggested that each SD increase in selenium increased the risk of MDD by 3% (OR_cIVW_ 1.03; 95% CIs 1.01–1.05; *p* = 0.001) and the risk of rMDD by 8% (OR_cIVW_ 1.08; 95% CI 1.02–1.15; *p* = 0.01). If this is a true estimate, the outcomes of the MoodFood trial could have been affected by the inclusion of selenium in its supplementation arm or by targeting selenium-deficient participants [16]. However, our results need to be interpreted cautiously. Firstly, although the confidence intervals are wide, cEgger point estimates are directionally opposite, suggesting a protective effect (OR_cEgger_ 0.94; 95% CI 0.84, 1.05; *p*= 0.29 for MDD; and OR_cEgger_ 0.97; 95% CI 0.67, 1.39; *p*= 0.86). Secondly, there is a lack of clarity over the functional role of the genetic variants with respect to selenium. The two lead genes in this region, DMGDH (dimethylglycine dehydrogenase) and BHMT2 (betaine homocysteine methytransferase), are hypothesised to be involved in selenoaminoacid metabolism, but a clear physiological pathway has yet to be defined. Finally, the relationship between selenium and MDD may be non-linear, with mood impacted by both deficiency and excess [81], which two-sample MR is unable to detect. Further work is required to establish the physiological underpinnings of the MR results.

We believe this is the first MR study to consider the effect of copper on MDD. Low dietary copper intake has been associated with MDD in observational studies [82], while circulating copper may be higher among depressed individuals [83]. Copper plays a role in direct neurological functions–such as sleep and memory–as well as having immunological effects, which could both hypothetically mediate a causal effect of copper on MDD [84]. However, effect sizes were small for both MDD (OR_cIVW_ 0.98; 95% CI 0.98–0.99; *p* = 0.0003) and rMDD (OR_cIVW_ 0.97; 95% CI 0.95–0.99; *p* = 0.002). As the small number of instruments for copper were derived from a single population, the estimates of the SNP-exposure association may have biased results. Furthermore, as the SNPs in the analyses had unclear physiological relevance for copper metabolism, further research is needed.

Aside from the inherent limitations of MR analyses, which are reviewed in depth elsewhere [18], a major limitation of this study is the lack of genetic data for many micronutrient exposures. Some exposures hypothetically linked to MDD were excluded from MR analyses, either due to the absence of GWAS studies (such as vitamin B3) or robustly associated instruments (such as chromium), meaning we were unable to corroborate reports of beneficial effects on depression [85,86]. Underpowered micronutrient GWAS’s affected the precision around some estimates, making it hard to rule out clinically important effects, particularly in rMDD. Furthermore, small exposure GWAS’s, may have missed some important genetic variants underlying biologically relevant pathways, impacting the validity of MR estimates. The data limitations meant we were unable to rule out small causal effects for any micronutrients, especially if the MR estimates are masking a larger threshold or U-shaped relationship. The limited number of available instruments often prevented sensitivity analyses, and some analyses were based on instruments lacking external validation in the original GWAS. In two-sample MR, weak instruments will generally bias the results towards the null, so as stronger instruments evolve from larger GWAS studies, replication and validation of these findings are warranted, along with one-sample MR allowing the exploration of non-linear effects. Where data availability allowed, we used non-traditional MR methods that account for correlated SNPs, such as cIVW and cEgger, to increase the power to detect small effects. Using these methods tightened the confidence intervals around our estimates, which otherwise all included the possibility of no effect. Effect sizes for MDD were generally small, and future trials should ensure adequate power to detect effects of these magnitudes, though the practicalities and clinical utility of primary prevention trials may be debatable. The direction of several of the point estimates suggested that increasing micronutrient status might increase the risk of MDD, highlighting the potential complexities of universal supplementation. Micronutrient prevention trials for MDD have not yielded the expected benefits [16,17] and, in some fields of medicine, have led to adverse effects [77], which could potentially be avoided by targeting interventions among participants with suspected or confirmed micronutrient deficiencies.

## 5. Conclusions

This large two-sample MR study of multiple micronutrients suggests tentative evidence for a protective effect of vitamin D, copper, and iron and a possible adverse effect of high selenium and magnesium on depressive outcomes. None of the analyses were resoundingly conclusive, with inconsistencies and caveats, and further clarification of the biological mechanisms involved is required.

Our analyses were well-powered to rule out large causal effects of the majority of micronutrients on depression. Although the magnitude of effects suggested here is modest at most, micronutrients may still contribute to the global disease burden of MDD given its high prevalence, recurrence, and morbidity and analyses in MDD and rMDD should be repeated as genetic samples evolve. Further, micronutrient MR studies using other psychiatric outcomes and in populations of other ancestry would be beneficial to highlight potential similarities and differences. Researchers planning future randomised trials of micronutrient supplementation should ensure adequate power given the likely small effect sizes and be cautious about increasing physiological levels, as for some micronutrients, this may be harmful.

## Figures and Tables

**Figure 1 nutrients-16-03690-f001:**
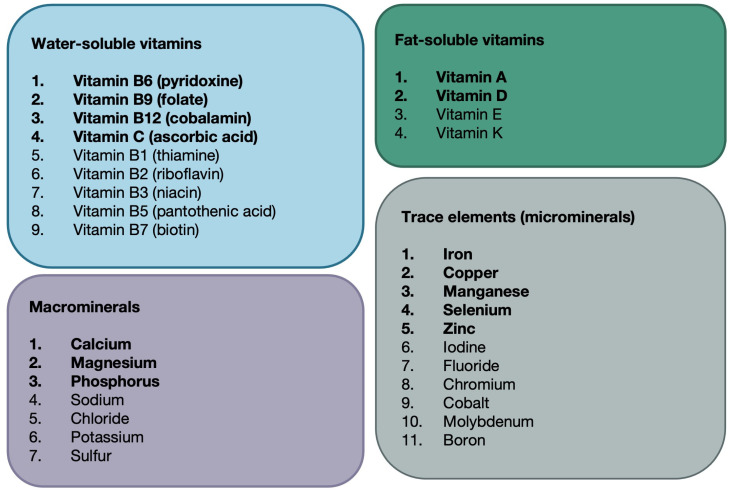
Micronutrient Classification. Micronutrients are essential components of the diet that are required in small amounts. For this analysis, we defined micronutrients as vitamins and minerals. We classified 31 micronutrients into four categories: water-soluble vitamins (n = 9), fat-soluble vitamins (n = 4), macrominerals (n = 7) and trace elements (n = 11). Micronutrients included in analyses are highlighted in bold, with reasons for exclusion highlighted in a flowchart (Figure 2).

**Figure 2 nutrients-16-03690-f002:**
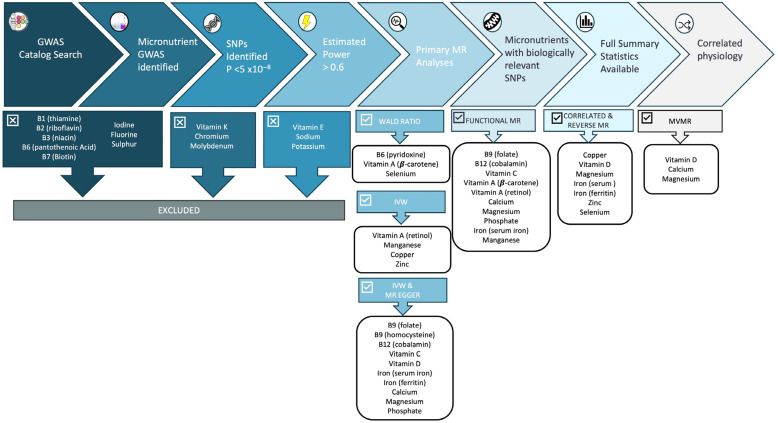
Flowchart of micronutrient selection and analyses. This flowchart explains the process of micronutrient selection and exclusion for these analyses, along with the MR methods used for each. Of the 31 micronutrients identified, fourteen were suitable for MR analyses: four water-soluble vitamins (vitamins B6, B9, B12 and C); two fat-soluble vitamins (vitamins A and D); three macrominerals (calcium, magnesium and phosphate); and five trace elements (iron, copper, manganese, selenium and zinc). For included micronutrients, the selection of MR methods depended on the number of SNPs available for each exposure, whether these SNPs mapped to genetic regions with a known physiological role in the synthesis, degradation or transport of the micronutrient (termed “functional analyses”), and whether full genetic summary statistics were available to facilitate additional analyses such as correlated MR methods (i.e., cIVW/cEgger), MVMR and reverse MR.

**Figure 3 nutrients-16-03690-f003:**
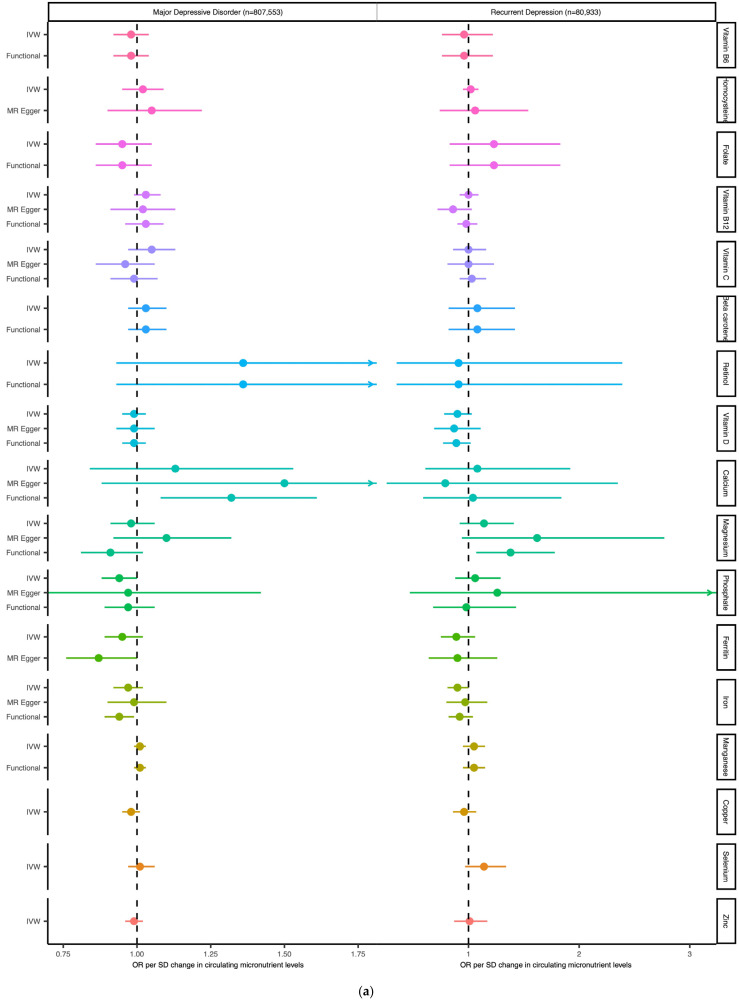
Forest plot of MR results for circulating micronutrients on the odds of MDD and rMDD. (**a**) MR estimated the Effects of Micronutrient exposures on MDD and rMDD, using traditional clumping thresholds (r^2^ < 0.001). (**b**) MR estimated the effects of micronutrient exposures using the full summary of available statistics. Comparing IVW and Egger estimates using traditional methods with correlated MR (cIVW and cEgger) and MVMR estimates.

**Table 1 nutrients-16-03690-t001:** Genetic instrument selection. Genetic instruments were selected from the largest available GWASs, apart from 25(OH)vitamin D, for which the largest GWAS was undertaken within UK Biobank [33], and so had complete sample overlap with our secondary outcome (rMDD) [34]. Columns include: ‘N’, GWAS sample size; ‘nSNPs’, number of SNPs in primary, correlated and functional analyses; ‘mFc’, mean F statistics; ‘R^2^ (%)’, estimated variance explained in exposure; ‘Power’, the estimated power to detect a 10% change in the odds of depression (see below); ‘SD’, standard deviation in natural units; ‘Natural units’ of GWAS; ‘Full data’, availability of full summary statistics for further analyses; ‘Replicated’, whether SNP-exposure association had been validated in a separate sample; ‘Converted’, whether data was converted into SD units from original publication. Exposures lacking nSNP data are those excluded due to power. Data for mFc, R2, and Power relate to primary analyses using traditional clumping thresholds, except for nSNPs, which report SNPs for correlated analyses in brackets where applicable. Publications in natural units were converted to SD units using the betas, and SE was converted using the transformation factor stated in the ‘SD’ (denoted by a * in the ‘Conv’ column). In some instances, SD was calculated from prior publications as unavailable in the original publication or from contacting corresponding authors. The ‘Power’ column provides an approximation of the power to detect a 10% change in the odds of depression (i.e., OR < 0.9; or >1.1) in MDD, assuming α = 0.05. Estimated power calculations should be treated cautiously as they have been derived using an online power calculator intended for one-sample, not two-sample MR (https://shiny.cnsgenomics.com/mRnd/ accessed on 15 January 2024).

	Exposure	Measure	N	nSNPs (Primary)	nSNPs (cIVW)	nSNPs (Func.)	mFc	R^2^ (%)	Power	SD	Natural Units	Full Data	Replicated	Converted
Water soluble Vitamins	Vitamin B6	Serum B6 [35]	1864	1		0	27	1.5	0.95	8.5	ng/mL		*	*
	Vitamin B9	Serum Folate [36]	~37,300	2		2	620	0.6	0.62	12.3	mmol/L		*	*
		Serum Homocysteine [37]	44,147	11		NA	101	2.8	1	82.2	µmol/L		*	
	Vitamin B12	Serum B12 [36]	45,576	9		7	192	5.4	1	140.0	pmol/L		*	*
	Vitamin C	Plasma Vitamin C [38]	52,018	11		1	85	1.8	0.98	20.1	umol/L			
Fat soluble vitamins	Vitamin A	Serum Retinol [39]	9302	1		2	56	2.4	0.99	301.0	μg/L		*	
		Beta Carotene [40]	~1800	2		1	99	1.1	0.86	0.3	umol/L		*	*
	Vitamin D	Serum 25(OH)D [34]	79,366	5	29	4	199	3.6	1	~31.9	nmol/L	*	*	
	Vitamin E	Serum alpha-tocopherol [41]	7781	-		NA	36	0.4	0.46	5.1	mg/L		*	
Macrominerals	Calcium	Serum Calcium [42]	~61,000	7		1	80	0.80	0.74	0.5	mg/dL		*	
	Magnesium	Serum Magnesium [43]	15,366	5	7	1	60	2.2	0.99	0.1	mmol/L	*	*	*
	Phosphate	Serum Phosphorus [44]	16,264	5		1	40	1.2	0.89	0.5	mg/dL		*	*
	Potassium	Urinary Potassium [45]	446,237	-		NA	45	0.0	0.09	33.8	mmol/L	*	*	
	Sodium	Urinary Sodium [45]	446,237	-		NA	48	0.2	0.24	44.4	mmol/L	*	*	
Trace elements	Iron	Serum Iron [46]	48,972	6	31	4	69	1.2	0.89	6.3	µmol/L	*	*	
		Serum Ferritin [46]		5	6	NA	324	4.3	1	123.3	μg/L	*	*	
	Copper	Erythrocyte Copper [47]	2603	2	5	0	62	5.3	1	~60	μg/L	*		
	Manganese	Serum Manganese [48]	949	2		2	50	11.0	1	37.0	mmol/L			
	Selenium	Serum Selenium [47]	5477	1	6	0	119	2.5	1	~219.8	μg/L	*	*	
	Zinc	Erythrocyte Zinc [47]	2603	2	3	0	61	4.2	1	~10	μg/L	*		

**Table 2 nutrients-16-03690-t002:** Results from two-sample MR in MDD (N = 430,775) and rMDD samples (N = 80,933).

				Major Depressive Disorder	Recurrent Depression			
	Exposure	Measure	Method	nSNP	OR (95% CI)	*p*	OR (95% CI)	*p*
Water Soluble Vitamins	Vitamin B6	Serum B6	Wald ratio	1	0.98 (0.92, 1.04)	0.45	0.96 (0.76, 1.22)	0.77
			Functional	-	No SNPs	-	-	-
	Vitamin B9	Homocysteine	IVW	12	1.02 (0.95, 1.09)	0.57	1.02 (0.87, 1.20)	0.81
			MR Egger	12	1.05 (0.90, 1.22)	0.56	1.06 (0.74, 1.54)	0.75
			Functional	-	Not applicable	-	-	-
		Serum Folate	IVW	2	0.95 (0.86, 1.05)	0.33	1.23 (0.83, 1.83)	0.31
			Functional	2	As above	-	-	-
	Vitamin B12	Serum B12	IVW	9	1.03 (0.99, 1.08)	0.15	1.00 (0.92, 1.09)	0.94
			MR Egger	9	1.02 (0.91, 1.13)	0.77	0.86 (0.72, 1.03)	0.14
			Functional	7	1.02 (0.94–1.11)	0.59	0.97 (0.89–1.06)	0.50
	Vitamin C	Plasma vitamin C	IVW	10	1.05 (0.96, 1.13)	0.24	1.00 (0.86–1.17)	1.00
			MR Egger	10	0.96 (0.86, 1.06)	0.44	1.00 (0.80, 1.25)	0.99
			Functional	1	0.99 (0.91, 1.07)	0.73	1.03 (0.92, 1.16)	0.60
Fat Soluble Vitamins	Vitamin A	Beta Carotene	Wald ratio	1	1.03 (0.97, 1.10)	0.34	1.08 (0.82, 1.42)	0.58
			Functional	1	As above	-	-	-
		Serum Retinol	IVW	2	1.36 (0.93, 1.99)	0.11	0.91 (0.35, 2.39)	0.85
			Functional	2	As above	-	-	-
	Vitamin D	Serum 25(OH)D	IVW	6	0.99 (0.95, 1.03)	0.56	0.90 (0.78, 1.03)	0.13
			MR Egger	6	0.99 (0.93, 1.06)	0.86	0.87 (0.67, 1.12)	0.36
			Functional	4	0.99 (0.95–1.03)	0.52	0.89 (0.77–1.02)	0.10
Macrominerals	Calcium	Serum Calcium	IVW	7	1.13 (0.84–1.53)	0.43	1.08 (0.61–1.92)	0.79
			MR Egger	7	1.50 (0.88–2.54)	0.2	0.79 (0.26–2.35)	0.69
			Functional	1	1.32 (1.08–1.61)	0.01	1.04 (0.59–1.84)	0.89
	Magnesium	Serum Magnesium	IVW	5	0.98 (0.91, 1.05)	0.56	1.14 (0.92, 1.41)	0.23
			MR Egger	5	1.10 (0.92, 1.32)	0.37	1.62 (0.94, 2.77)	0.18
			Functional	1	0.91 (0.81–1.02)	0.11	1.38 (1.07–1.78)	0.01
	Phosphate	Serum phosphorous	IVW	5	0.94 (0.88, 1.00)	0.04	1.06 (0.88, 1.29)	0.53
			MR Egger	5	0.97 (0.66, 1.42)	0.88	1.26 (0.47, 3.36)	0.68
			Functional	1	0.97 (0.89–1.06)	0.52	0.98 (0.68–1.43)	0.92
Microminerals	Copper	Erythrocyte copper	IVW	2	0.98 (0.95, 1.01)	0.19	0.96 (0.86, 1.07)	0.46
			Functional	-	No SNPs	-	-	-
	Iron	Serum Ferritin	IVW	5	0.95 (0.89, 1.02)	0.15	0.89 (0.75, 1.06)	0.20
			MR Egger	5	0.87 (0.76, 1.00)	0.14	0.90 (0.64, 1.26)	0.58
			Functional	-	Not applicable	-	-	-
		Serum Iron	IVW	4	0.98 (0.93, 1.02)	0.31	0.90 (0.81, 1.00)	0.05
			MR Egger	4	0.99 (0.90, 1.10)	0.93	0.97 (0.80, 1.17)	0.76
			Functional	4	0.94 (0.89, 0.99)	0.02	0.92 (0.82, 1.04)	0.18
	Manganese	Whole blood manganese	IVW	2	1.01 (0.99, 1.03)	0.42	1.05 (0.95, 1.15)	0.38
			Functional	2	As above	-	-	-
	Selenium	Erythrocyte selenium	Wald ratio	1	1.01 (0.97, 1.06)	0.65	1.14 (0.97, 1.34)	0.12
			Functional	-	No SNPs	-	-	-
	Zinc	Erythrocyte zinc	IVW	2	0.99 (0.96, 1.02)	0.57	1.01 (0.87, 1.17)	0.87
			Functional	-	No SNPs	-	-	-

**Table 3 nutrients-16-03690-t003:** Results for analyses using correlated MR methods (cIVW and cEgger).

	Exposure	Method	Clump r^2^	nSNP	Major Depressive Disorder (n = 430,775) OR (95% CI), *p* Value		Recurrent Depression (n = 80,933) OR (95% CI), *p* Value	
Vitamins	Vitamin D	cIVW	0.2	29	0.99 (0.93, 1.06)	0.86	0.87 (0.74, 1.02)	0.08
		cEgger	0.2	29	0.86 (0.78, 0.95)	0.003	0.81 (0.64, 1.03)	0.09
		Traditional IVW	0.001	6	0.99 (0.95, 1.03)	0.56	0.90 (0.78, 1.03)	0.13
		Traditional Egger	0.001	6	0.99 (0.93, 1.06)	0.86	0.87 (0.67, 1.12)	0.36
Macrominerals	Magnesium	cIVW	0.2	7	1.01 (0.95, 1.07)	0.79	1.12 (0.95, 1.33)	0.19
		cEgger	0.2	7	1.07 (0.86, 1.34)	0.53	1.61 (0.97, 2.68)	0.07
		Traditional IVW	0.001	5	0.98 (0.91, 1.05)	0.56	1.14 (0.92, 1.41)	0.23
		Traditional Egger	0.001	5	1.10 (0.92, 1.32)	0.37	1.62 (0.94, 2.77)	0.18
Microminerals	Iron	cIVW	0.2	31	0.97 (0.94, 1.00)	0.03	0.87 (0.83, 0.91)	3.2 × 10^−9^
		cEgger	0.2	31	0.98 (0.92, 1.04)	0.47	0.88 (0.80, 0.97)	0.01
		Traditional IVW	0.001	4	0.98 (0.93, 1.02)	0.31	0.90 (0.81, 1.00)	0.05
		Traditional Egger	0.001	4	0.99 (0.90, 1.10)	0.93	0.97 (0.80, 1.17)	0.76
	Ferritin	cIVW	0.2	6	0.97 (0.9, 1.04)	0.39	0.89 (0.75, 1.06)	0.20
		cEgger	0.2	6	0.90 (0.79, 1.02)	0.11	0.87 (0.67, 1.14)	0.31
		Traditional IVW	0.001	5	0.95 (0.89–1.02)	0.15	0.89 (0.75–1.06)	0.20
		Traditional Egger	0.001	5	0.87 (0.76–1.00)	0.14	0.90 (0.64–1.26)	0.58
	Copper	cIVW	0.2	7	0.98 (0.98, 0.99)	0.0003	0.97 (0.95, 0.99)	0.002
		cEgger	0.2	7	0.98 (0.93, 1.04)	0.58	1.00 (0.91, 1.10)	1
		Traditional IVW	0.001	2	0.98 (0.95, 1.01)	0.19	0.96 (0.86, 1.07)	0.46
		Traditional Egger	0.001	2	Insufficient SNPs	-	Insufficient SNPs	-
	Selenium	cIVW	0.2	6	1.03 (1.01, 1.05)	0.001	1.08 (1.02, 1.15)	0.01
		cEgger	0.2	6	0.94 (0.84, 1.05)	0.29	0.97 (0.67, 1.39)	0.86
		Traditional IVW	0.001	1	1.01 (0.97, 1.06)	0.65	1.14 (0.97, 1.34)	0.12
		Traditional Egger	0.001	1	Insufficient SNPs	-	Insufficient SNPs	-
	Zinc	cIVW	0.2	7	1.00 (0.98, 1.02)	0.74	1.04 (1.00, 1.07)	0.03
		cEgger	0.2	7	1.01 (0.96, 1.05)	0.72	1.08 (1.01, 1.15)	0.02
		Traditional IVW	0.001	2	0.99 (0.96, 1.02)	0.57	1.01 (0.87, 1.17)	0.87
		Traditional Egger	0.001	2	Insufficient SNPs	-	Insufficient SNPs	-

MR estimates for micronutrients using correlated IVW (cIVW) and correlated MR Egger (cEgger) methods, where full summary statistics were available. Results are given as OR per SD increase in exposure, with 95% CI. These estimates use a relaxed clumping threshold (r^2^ < 0.2) to increase the number of SNPs in each analysis. The analysis accounts for LD between SNPs to increase analytical power without double counting the estimates. Heterogeneity statistics for these analyses are presented in Appendix A.

**Table 4 nutrients-16-03690-t004:** Multivariable MR results.

				Major Depressive Disorder (n = 430,775)		Recurrent Depression (n = 80,933)			
		nSNPs	F	OR per SD (95% CI)	*p*	OR per SD (95% CI)	*p*	MDD Q (*p*)	rMDD Q (*p*)
Magnesium	Unadjusted	5	60	0.98 (0.91–1.06)	0.60	1.14 (0.92–1.41)	0.23	9 (0.07)	7 (0.15)
	Calcium	10	29	0.98 (0.90–1.07)	0.64	0.87 (0.68–1.12)	0.32	21 (0.004)	15 (0.03)
	Vitamin D	10	28	0.97 (0.92–1.04)	0.44	0.88 (0.72–1.07)	0.24	11 (0.13)	9 (0.23)
	Both	15	19	0.98 (0.91–1.06)	0.60	0.88 (0.70–1.09)	0.26	25 (0.01)	18 (0.09)
Vitamin D	Unadjusted	6	199	0.99 (0.95–1.03)	0.56	0.90 (0.78–1.03)	0.13	2 (0.80)	2 (0.79)
	Calcium	10	212	0.99 (0.87–1.13)	0.86	0.75 (0.48–1.17)	0.24	13 (0.06)	11 (0.15)
	Magnesium	10	197	1.01 (0.89–1.14)	0.90	0.75 (0.49–1.14)	0.22	11 (0.13)	9 (0.23)
	Both	15	135	0.99 (0.86–1.15)	0.92	0.75 (0.47–1.2)	0.26	25 (0.01)	18 (0.09)
Calcium	Unadjusted	7	80	1.13 (0.84–1.53)	0.43	1.08 (0.61–1.92)	0.79	23 (0.001)	9 (0.17)
	Magnesium	10	174	1.09 (0.98–1.21)	0.17	1.01 (0.77–1.32)	0.96	21 (0.004)	15 (0.03)
	Vitamin D	10	365	1.09 (1.01–1.19)	0.07	1.02 (0.82–1.27)	0.86	13 (0.06)	11 (0.15)
	Both	15	118	1.07 (0.98–1.18)	0.15	1.01 (0.80–1.27)	0.94	25 (0.01)	18 (0.09)

Effect estimates using MVMR to account for correlated physiological effects of vitamin D, calcium and magnesium did not alter univariable MR estimates for MDD. However, the MVMR estimated effect of serum magnesium on rMDD was in the reverse direction compared to univariable estimates, suggesting more complex physiology may be driving the functional and correlated MR effect estimates. Unadjusted estimates are given as IVW results, with mean F statistics referring to conditional F statistics in MVMR models. NB MVMR models, including calcium, may be affected by sample overlap with recurrent depression.

**Table 5 nutrients-16-03690-t005:** Reverse MR analyses.

Micronutrient	Method	nSNP	Unfiltered OR (95% CIs)	*p*	nSNP	Steiger Filtered OR (95% CIs)	*p*
Vitamin D	Inverse variance weighted	66	−0.02 (−0.05–0.01)	0.15	53	0.00 (−0.02–0.02)	0.72
	MR Egger	66	0.00 (−0.16–0.16)	0.96	53	−0.05 (−0.19–0.09)	0.51
	cIVW	110	−0.02 (−0.04–0.00)	0.09	91	0.00 (−0.01–0.02)	0.67
	cEgger	110	0.06 (−0.07, 0.19)	0.34	91	0.01 (−0.09, 0.12)	0.80
Magnesium	Inverse variance weighted	42	−0.04 (−0.19–0.11)	0.62	26	−0.02 (−0.21–0.17)	0.87
	MR Egger	42	0.36 (−0.43–1.15)	0.37	26	−0.26 (−1.30–0.78)	0.63
	cIVW	60	−0.05 (−0.19–0.08)	0.42	36	0.01 (−0.16–0.17)	0.94
	cEgger	60	0.28 (−0.41, 0.96)	0.43	36	−0.10 (−0.97, 0.78)	0.83
Copper	Inverse variance weighted	66	0.04 (−0.24–0.32)	0.77	15	0.07 (−0.55–0.69)	0.82
	MR Egger	66	0.15 (−1.49–1.79)	0.86	15	0.85 (−4.42–6.12)	0.76
	cIVW	110	0.16 (−0.05–0.37)	0.14	27	0.03 (−0.43–0.48)	0.91
	cEgger	110	0.75 (−0.68, 2.18)	0.31	27	0.51 (−2.74, 3.75)	0.76
Iron	Inverse variance weighted	64	0.02 (−0.08–0.12)	0.65	46	−0.02 (−0.14–0.1)	0.79
	MR Egger	64	0.44 (−0.19–1.07)	0.18	46	−0.16 (−0.98–0.66)	0.71
	cIVW	107	0.05 (−0.04–0.13)	0.13	71	−0.01 (−0.11–0.08)	0.08
	cEgger	107	0.37 (−0.16, 0.91)	0.17	71	−0.08 (−0.66, 0.50)	0.79
Ferritin	Inverse variance weighted	64	0.00 (−0.10–0.10)	0.980	48	0.00 (−0.11–0.11)	0.98
	MR Egger	64	0.62 (0.03–1.21)	0.05	48	0.16 (−0.62–0.94)	0.68
	cIVW	104	0.00 (−0.09, 0.09)	0.98	67	0.02 (−0.08, 0.11)	0.74
	cEgger	104	0.24 (−0.29, 0.78)	0.37	67	0.02 (−0.61, 0.65)	0.95
Selenium	Inverse variance weighted	66	0.01 (−0.29–0.31)	0.81	18	0.00 (−0.58–0.58)	0.99
	MR Egger	66	0.88 (−0.90–2.66)	0.32	18	0.50 (−4.59–5.59)	0.85
	cIVW	110	0.27 (0.01–0.52)	0.04	29	0.07 (−0.37–0.51)	0.77
	cEgger	110	0.51 (−1.17, 2.20)	0.55	29	0.69 (−2.99, 4.36)	0.71

For micronutrients with evidence suggestive of a causal effect, we undertook to reverse MR, using SNPs identified from the complete PGC MDD sample (n = 807,553) as proxies for MDD and the micronutrient as outcomes. The results shown are before (left) and after (right) Steiger filtering to remove the SNPs more strongly associated with the outcome (i.e., the micronutrient). Clumping thresholds are as in forward analyses (r^2^ < 0.001 for traditional MR and r^2^ < 0.2 for correlated methods.) Heterogeneity statistics for reverse MR are provided in Appendix A.

## Data Availability

Further information about obtaining access to the PGC summary statistics is available from http://www.med.unc.edu/pgc/statgen. The full GWAS summary statistics for the 23andMe discovery data set will be made available through 23andMe to qualified researchers under an agreement with 23andMe that protects the privacy of the 23andMe participants. Please visit https://research.23andme.com/collaborate/#dataset-access/for more information and to apply to access the data.

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
