# Peer review of "Micronutrients and Major Depression: A Mendelian Randomisation Study"

_nutrients, 2024, doi:10.3390/nu16213690_

Round 1

Reviewer 1 Report

Comments and Suggestions for Authors

First, I would like to commend the authors for undertaking this important work. While generally the protocol is well-written and structured, I have some comments that need to be addressed. My comments are directed in the following key sections Nutrients publication criteria:

*       Abstract and Plain Language Summary
*       Background section of the review
*       Methods
*       Intervention
*       Discussion

Abstract and plain language summary (Main text body)

1. Briefly define what MR  is to the general public reader

Background

This is well written no comment

Methods

1.     Can Figure 2, Flowchart of micronutrient selection and analyses be improved hard for a reader to follow through it is blurred

2.     Revise table 1 it is too crowded for the readers retain what is very relevant

3.     Figure 3. Forest plot of MR results for circulating micronutrients on the odds of MDD  and rMDD is hardly clear to read

4.     Revise table 5 so that the Odd ratios are in the  same row

Discussion

5.     Any reason why the authors used the large number of SNPs with uncertain relevance to vitamin D physiology?

6.     The authors themselves mention that the results need to be interpreted cautiously, as evidenced by very confidence intervals, thus is their conclusion reliable for public health prevention of depression?

7.     Given the findings of the study, are there any plans for the authors to have another study with different ancestry?

Author Response

Dear Reviewer 1,

Thank you for giving us the opportunity to improve our manuscript, by taking the time to read and consider it.

Please find the specific responses to your comments and questions below.

Many thanks,

Rebecca Carnegie et al.

Abstract and plain language summary (Main text body)

  1. Briefly define what MR  is to the general public reader 

Thank you. We have amended the abstract to include:

Line 31                  “To estimate the effects of micronutrient exposures on major depressive disorder (MDD) and recurrent depression (rMDD) using Mendelian randomization (MR), a method using genetic data to estimate causal effects given certain assumptions.”

Methods

  1. Can Figure2, Flowchart of micronutrient selection and analyses be improved hard for a reader to follow through it is blurred

Thank you for this suggestion, we agree it is overly complicated. We have included a new figure hoping to improve the readability. We have also updated table 1 to fit with this.

  1. Revise table 1 it is too crowded for the readers retain what is very relevant

Thank you. We have simplified table 1 to reduce visual clutter, such as removing PMID/ author details. We hope this improves readability whilst maintaining required data.

  1. Figure 3. Forest plot of MR results for circulating micronutrients on the odds of MDD  and rMDD is hardly clear to read

The image quality of figure 3 has been improved to improve acuity.

  1. Revise table 5 so that the Odd ratios are in the  same row

Thank you, this has been amended.

Discussion

  1. Any reason why the authors used the large number of SNPs with uncertain relevance to vitamin D physiology?

We assume this comment is referring to the paper by Revez et al, in which MR analyses were appended to a UK Biobank GWAS of 25(OH)D. Since data availability has increased, it has become common for GWAS studies to undertake a hypothesis free approach to MR using multiple outcomes in appendices or supplementary material. In these instances, it is common practice to use all SNPs reaching genome wide significance thresholds, which may quickly identify any metaphorical ’low lying fruit’. Both approaches are to MR analyses are valid, and each has limitations, so comparing estimates using different approaches can be informative.

  1. The authors themselves mention that the results need to be interpreted cautiously, as evidenced by veryconfidence intervals, thus is their conclusion reliable for public health prevention of depression?

We have been cautious not to over-interpret the findings in this paper, as MR results alone should not be read in isolation but considered more broadly in the context of other methodologies, such as intervention trials. It would be premature to advise on public health prevention strategies at this point. However, the results can highlight which micronutrients might be the most important candidates for depression prevention trials, to inform sample size estimates to ensure sufficient statistical power, and also may prompt mechanistic research into pathophysiological pathways between micronutrient exposures and depression, such as using two-step approaches.

  1. Given the findings of the study, are there any plans for the authors to have another study with different ancestry?

We have considered repeating analyses using different ancestries, although micronutrient exposure data is already limited among European ancestry populations, and may be even more so among other ancestries. It may be necessary to await improvements in micronutrient GWAS datasets. We would be happy to collaborate on future projects if this is of interest.

Reviewer 2 Report

Comments and Suggestions for Authors

Comment. It seems that adding a discussion of findings from prospective cohort studies and vitamin D supplement studies as well as mechanisms identified would strengthen the manuscript. Suggest searching Google Scholar for recent articles and reviews for vitamin D and copper (the manuscript mentioned observational studies regarding iron). A quick search suggests that vitamin D and iron are beneficial but copper is not.

Prospective associations between vitamin D and depression in middle-aged adults: Findings from the UK Biobank cohort

A Ronaldson, JA de la Torre, F Gaughran… - Psychological …, 2022 - cambridge.org

Molecular basis underlying the therapeutic potential of vitamin D for the treatment of depression and anxiety

BR Kouba, A CamargoJ Gil-Mohapel… - International journal of …, 2022 - mdpi.com

Copper in depressive disorder: A systematic review and meta-analysis of observational studies

M Ni, Y You, J Chen, L Zhang - Psychiatry research, 2018 - Elsevier

As far as we are aware, no studies have considered the effect of 25(OH) 615 vitamin D on the risk of recurrent depression.

Comment: From a search of Google Scholar:

Analysis of vitamin D status in major depression

…, A Wozniacka, D Strzelecki - Journal of Psychiatric …, 2014 - journals.lww.com

… with depression. Our finding of low serum levels of vitamin D in patients with recurrent
depression suggests that these patients represent an important group that is at risk for vitamin D …

Plasma vitamin D status and recurrent depressive symptoms in the French SU. VI. MAX cohort

C Collin, KE Assmann, M Deschasaux… - European journal of …, 2017 - Springer

Prospective associations between vitamin D and depression in middle-aged adults: Findings from the UK Biobank cohort

A Ronaldson, JA de la Torre, F Gaughran… - Psychological …, 2022 - cambridge.org

893 36. Coleman JRI, Gaspar HA, Bryois J, Breen G. The genetics of the mood disorder spectrum: genome-wide

894 association analyses of over 185,000 cases and 439,000 controls. Biol Psychiat

Comment: Incomplete reference.

Biol Psychiatry2020 Jul 15;88(2):169-184.

 doi: 10.1016/j.biopsych.2019.10.015.

The limitations of MR analyses should be discussed in more detail. One important limitation is that generally only a few SNPs for any relationship are used. As a result, the SNPs not used may be more important than the ones used. I suggest searching Google Scholar with “limitations, mendelian randomization studies, nutrients"

Comments on the Quality of English Language

The English language is good.

Author Response

Dear Reviewer 2,

Thank you for giving us the opportunity to improve our manuscript, by taking the time to read and consider it.

Please find the specific responses to your comments and questions below.

Many thanks,

Rebecca Carnegie et al.

Comments and Suggestions for Authors

  1. It seems that adding a discussion of findings from prospective cohort studies and vitamin D supplement studies as well as mechanisms identified would strengthen the manuscript. Suggest searching Google Scholar for recent articles and reviews for vitamin D and copper (the manuscript mentioned observational studies regarding iron). A quick search suggests that vitamin D and iron are beneficial but copper is not.

Thank you for this suggestion, we agree, the discussion is improved by further reference to observational and interventional studies. We hope the adaptations are acceptable, given limitations in word count.

Specific additions include:

797-799               “Vitamin D has been the most thoroughly investigated of all micronutrients in depression, with inverse associations in observational studies,62-64 although mixed results among trials.65,66

834-5   “…evidence from observational studies suggesting an increased risk of depression among those with iron deficiency anaemia, reduced by supplementation.67,68”

902-5   Low dietary copper intake has been associated with MDD in observational studies,79 while circulating copper may be higher among depressed individuals.80”

  1. “As far as we are aware, no studies have considered the effect of 25(OH) 615 vitamin D on the risk of recurrent depression.”

Thank you for pointing this out. We have amended the sentence to specify MR studies, as we concur it was unintentionally vague and could be misleading.

817         “As far as we are aware, no MR studies have considered the effect of 25(OH) vitamin D on the risk of recurrent depression.”

  1. Comment: Incomplete reference.

Thank you, this has been amended.

  1. The limitations of MR analyses should be discussed in more detail. One important limitation is that generally only a few SNPs for any relationship are used. As a result, the SNPs not used may be more important than the ones used. I suggest searching Google Scholar with “limitations, mendelian randomization studies, nutrients"

Thank you for this insight. We have mentioned MR limitations in a few locations, particularly in the final discussion paragraph, as well as directing readers towards a more detailed review. However, we agree that clearly defining methodological limitations is important, and have added the following to further emphasize:

898        the relationship between selenium and MDD may be non-linear, with mood impacted by both deficiency and excess,78 which two-sample MR is unable to detect.”

924…small exposure GWAS’s, may have missed important genetic variants underlying biologically relevant pathways, impacting the validity of MR estimates.”

We hope these changes strike the right balance between providing enough detail without compromising the readability, which is already challenged by the complexity of multiple exposures.

Reviewer 3 Report

Comments and Suggestions for Authors

I congratulate the authors for this very huge work.

I have only 1 consideration for improve the paper:

1-Figure 3 increase the clarity of figure 3, the first scheme is not readeable.

Comments on the Quality of English Language

None

Author Response

Dear Reviewer 3,

Thank you for giving us the opportunity to improve our manuscript, by taking the time to read and consider it.

Please find the specific responses to your comments and questions below.

Many thanks,

Rebecca Carnegie et al.

Comments and Suggestions for Authors

I congratulate the authors for this very huge work. I have only 1 consideration for improve the paper:

  1. Figure 3 increase the clarity of figure 3, the first scheme is not readeable.

Thank you, we have amended figure 3a to improve the clarity.

Reviewer 4 Report

Comments and Suggestions for Authors

excellent research, only if its possible sinthetize resultast and methods becase are so large

Author Response

Dear Reviewer 4,

Thank you for giving us the opportunity to improve our manuscript, by taking the time to read and consider it.

Please find the specific responses to your comments and questions below.

Many thanks,

Rebecca Carnegie et al.

Comments and Suggestions for Authors

  1. excellent research, only if its possible synthetize results and methods because are so large

Thank you for your kind comments. We have amended figure 1 to try to improve the visualization of methods used. We have also tried to improve the image quality of the results shown in figure 3. We hope this makes the details clearer.